# Assessing a Community Partnership Addressing Food Security Among Older Adults During COVID-19

**DOI:** 10.3390/ijerph22020163

**Published:** 2025-01-26

**Authors:** Jenny Jinyoung Lee, Christy Nishita, Kathryn L. Braun

**Affiliations:** Thompson School of Social Work & Public Health, University of Hawai‘i at Mānoa, Honolulu, HI 96822, USA; cnishita@hawaii.edu (C.N.); kbraun@hawaii.edu (K.L.B.)

**Keywords:** food security, older adults, community partnerships, collective impact, COVID-19, emergency response, public health coalition

## Abstract

For many vulnerable older adults, food access was disrupted during the COVID-19 pandemic. In Hawai‘i, the Kūpuna (the Hawaiian word for elders) Food Security Coalition (KFSC) was formed in March 2020 to address this challenge, leveraging local and federal funding support. This case study presents information on coalition formation and success in addressing this emergency, as well as evaluation data on coalition functioning as assessed by the Collective Impact (CI) framework. Coalition functioning was assessed across the five CI conditions: common agenda, shared measurement, mutually reinforcing activities, continuous communication, and backbone support. Case study data were available from interview and learning circle transcripts, survey findings, and other program documents. Between March and December 2020, the KFSC coordinated efforts of 46 organizations to serve approximately 1.2 million meals to 8300 vulnerable seniors in Honolulu County. Within the first 9 months of existence, the coalition’s measurement system and the common agenda conditions showed advanced maturity, while the other conditions demonstrated moderate maturity levels. Despite challenging leadership transitions, the coalition was successful in helping increase food access and then pivoting in 2021 to promote kūpuna vaccinations, and the coalition continues to meet regularly to address issues of concern to vulnerable older adults. This study provides evidence-based guidance for communities seeking to establish public/non-profit partnerships for emergency food response for older adults, demonstrating how structured coalition approaches can effectively mobilize and coordinate multi-stakeholder efforts during and beyond crises.

## 1. Introduction

The COVID-19 pandemic severely disrupted meal programs and food access for vulnerable older adults when congregate meal sites closed, home-delivered meal services became strained, and the fear of infection limited their ability to go out and purchase food [1,2,3]. For many older adults, accessing adequate food was already challenging due to physical, mental, and social barriers associated with aging before the pandemic [4,5,6,7,8,9]. Pandemic-associated disruptions to food required rapid, coordinated responses from multiple stakeholders to address immediate needs that crossed economic, social, health, and environmental realms [10].

Coalitions and community partnerships have emerged as a preferred method to address complex social problems, such as environmental improvement, economic development, educational reforms, and public health [11,12,13,14,15]. In this context, a coalition refers to a group of diverse organizations and stakeholders from different sectors who come together with a common goal to work collaboratively to solve a problem. These types of partnerships are more effective than single organizations at addressing broad issues of community concern [12,16,17]. For example, Janosky et al. demonstrated the benefits of a multisector community coalition to address the overall health of the citizens of Summit County, Ohio, by increasing the effectiveness of existing resources [18].

The Collective Impact (CI) framework [19], introduced by Kania and Kramer, provides a structured approach for organizations to collaborate in reaching a common goal [10,20,21,22,23,24,25]. This framework emphasizes five key conditions essential for successful collaboration: a common agenda, shared measurement, mutually reinforcing activities, continuous communication, and backbone support [26]. Recent studies have demonstrated the framework’s effectiveness in guiding food systems initiatives [10,27], such as Michigan’s Good Food Charter [10] and South Dakota’s Local Foods Collaborative [27].

However, there is a lack of research on the success of this approach in addressing compromised food access among older adults. This gap is particularly significant in the context of emergency response, as older adults face unique vulnerabilities during crises that can exacerbate food inaccessibility [28]. The COVID-19 pandemic highlighted the need for a rapid, coordinated response to ensure food access for vulnerable populations [29], but there is limited evidence on how coalitions can effectively apply the CI approach in these urgent situations. The gap is particularly concerning given the rapid aging of many communities and the increasing frequency of natural disasters and public health emergencies that disproportionately affect older adults [30]. While existing research has shown the value of coalitions in addressing food systems and public health challenges [10,31,32], there is limited evidence on how to structure and sustain these efforts specifically for older adults’ food security, especially in crisis situations that require immediate response.

The Kūpuna (the Hawaiian word for elders) Food Security Coalition (KFSC) was established in March 2020 in Honolulu County to address disruptions in food services for older adults during COVID-19. The KFSC brought together 46 organizations serving older adults and/or engaged in food production, food distribution, and other resources (Figure 1). Their immediate goal was to increase access to food among kūpuna during the pandemic by ensuring access to nutritious meals and wrap-around services that support older adults to age in place [33], with the hope that the coalition could continue after the pandemic to address issues concerning older adults.

Conceptual models are essential for evaluating public health programs [16], with several found in the literature specific to assessing public health partnerships. For example, Janosky et al. used the Health Impact Pyramid to assess a coalition’s emergency response effort [18], while Cramer et al. introduced the Internal Coalition Outcome Hierarchy to promote public health collaboration [16]. Kegler et al. highlighted the Community Coalition Action Theory as a framework for healthy cities initiatives [34]. This study evaluated the KFSC’s work during its first 9 months of operation using the CI framework as a guideline to assess the KFSC’s successes, challenges, and lessons learned in increasing food access for vulnerable older adults during the COVID-19 emergency.

By providing an in-depth examination of the KFSC’s experience, this study contributes to the limited body of knowledge on the application of CI in addressing older adult food security, particularly in crisis situations. The findings offer valuable insights for other communities seeking to establish effective coalitions to support older adults during emergencies and beyond. Furthermore, by using the CI framework to guide the evaluation, this study explores the potential of CI as a tool for assessing and enhancing coalition-based interventions in the field of aging and food security.

## 2. Materials and Methods

### 2.1. Study Design

This evaluation followed a case study approach similar to Lynn et al. [35] to assess the KFSC’s work over the first nine months (March–December 2020) of operation. Case studies are used to gain a deep understanding of an issue or event within its real-life context [36,37]. This method was chosen to explore the KFSC’s actions and extract lessons for similar initiatives, given the limited research on coalition-based efforts to increase food access among vulnerable older adults during public health emergencies [38,39].

### 2.2. Data Collection and Analysis

Yin explained that case studies rely on multiple data sources to triangulate findings [37]. Following this guidance, several types of documents (interview and learning circle transcripts, survey findings, and management documents) were used to assess the five CI conditions to identify best practices and areas for improvement, as well as to inform an after-action report for funders (Table 1). To enhance the trustworthiness of the findings, this study employed several key strategies, including triangulation across multiple data sources and stakeholder perspectives, review of the preliminary findings with the KFSC leadership, and maintenance of a detailed audit trail documenting analytical decisions. These validation strategies align with established practices for ensuring rigor in qualitative organizational research [40,41].

#### 2.2.1. Data Sources

Transcripts of previously conducted semi-structured interviews (*n* = 5): Five one-on-one interviews were conducted with key coalition members between 22 October and 12 November 2020 by two members of the Data Committee and a staff member from the backbone organization. These interviews were aimed at gathering insights for an after-action report and involved stakeholders from various sectors (a government officer, a philanthropic funder, a service provider, a community organizer, and a policy advocate). The questions are shown in Table 1. Due to COVID-19 restrictions, four interviews were conducted via video meetings, and one interview was conducted via email. The video interviews were recorded, transcribed, and subsequently summarized by notetakers.

Transcripts of two previously conducted learning circle discussions (*n* = 37): Two virtual learning circle discussions were conducted in August 2020 and November 2020. These discussions were part of the grant requirements for organizations receiving KFSC funding to address food inaccessibility among vulnerable elders. The first learning circle included 14 participants from 10 organizations, and the second included 23 participants from 9 organizations. These organizations provided a range of services, including meal and food delivery, health and wellness checks, referrals, food program education, and grocery shopping and delivery for seniors unable to shop due to health or mobility issues. The questions are shown in Table 1. These sessions were transcribed by a notetaker.

Results of previously conducted surveys (*n* = 18): Two anonymous surveys were conducted, one in August and the other in November 2020. These were completed by representatives from the 18 agencies funded through the coalition to increase food access among vulnerable older adults. These surveys gathered feedback on the KFSC’s operations through multiple-choice and open-ended questions. The survey items are shown in Table 1.

Management documents: More than 1000 management documents, such as member lists, meeting notes, emails, and grant-related materials, were reviewed, primarily from 2020, along with some 2021 documents that summarized activities from the previous year. These management documents included all documents contained on the KFSC’s shared Google Drive dated 2020, as well as all group emails for the Steering Committee, Funding Committee, and Data Committee for that same period. Documents were collected by two of the authors of this study who were members of the Steering Committee of the KFSC and were provided access to the drives for the purposes of this study. The details are provided in Table 2. These documents were reviewed by the lead author of this study to identify roles, key dates, decisions, resource allocations, participation, engagement, successes, opportunities, and lessons learned. A deductive thematic analysis was used to code information from these documents under the CI framework, leveraging electronic categorization and search capabilities that allowed for targeted content analysis by members of the study team. The study team also assessed tracking reports for service delivery and data consistency.

#### 2.2.2. Analytical Framework

The CI framework, with its five CI conditions (common agenda, shared measurement, mutually reinforcing activities, continuous communication, and backbone support) was used to guide data analysis. A Collective Impact Assessment Rubric, developed by Lynn et al. [35] was used to assess how well each condition was operationalized. This rubric includes four criteria or elements for each condition to assess how well the condition is operationalized. For example, the four elements for the “common agenda” condition are (1) the presence of an identifiable, overarching goal and vision for the initiative within a clearly defined, actionable problem space; (2) partners’ shared understanding of the problem; (3) a clearly articulated approach or set of high-level strategies to solve the problem; and (4) partners’ high level of buy-in to the shared vision for change, agreed-upon goals, and approaches. Leveraging this rubric, a deductive approach was used to code data from the transcripts, documents, and surveys under each of the key conditions and elements to help determine the maturity and effectiveness of the coalition’s work in each condition (Table 3) [31]. The level of coalition maturity was determined by the number of elements met within each condition. For example, the coalition is considered to have a high level of maturity in operationalizing a condition when all four elements for that condition are present or met. Additionally, inductive coding was used to identify and label themes that did not fit the CI framework. Content analysis was used to assess KFSC processes month-by-month, along with 9-month outcomes, for example, the partners engaged, funds leveraged, meals served, and kūpuna reached.

## 3. Results

### 3.1. Description of the Kūpuna Food Security Coalition

The KFSC was initiated by the Elderly Affairs Division (EAD), the Area Agency on Aging for Honolulu County, in March 2020 to increase food access among vulnerable older adults in Honolulu County in response to the COVID-19 pandemic. Honolulu County has 1,000,000 residents, and approximately 240,000 of them are 60 or older. Initial partners included the AARP (formerly the American Association of Retired Persons) Hawaiʻi, Aloha United Way, the Harry and Jeanette Weinberg Foundation, and the University of Hawaiʻi Center on Aging. Forming the KFSC’s core team, these partners recruited other organizations through their networks, eventually involving 46 organizations from food providers, meal delivery services, and providers of support services to funders and government agencies. By becoming a member of the coalition, these organizations agreed to share resources and expertise, attend meetings, share best practices, and work together to increase food access for older adults.

The core team established guidelines, identified target populations and resources, and provided organizational support. In the first month, the KFSC members were organized into three subgroups focusing on food resources, funding, and access. While each of the individual members maintained their day-to-day operations, the roles and operational model (Figure 2) for the KFSC itself were developed by the subgroups. The access subgroup built and maintained a database of food resources that was leveraged by 211 Aloha United Way (a call center that provides older adults with information on resources), Kanu Hawaiʻi (a non-profit that provided the connection platform between service providers and volunteers), and the EAD call center to connect older adults in need with available food resources. The food subgroup was coordinated across members to ensure food availability and delivery. The funding subgroup, led by the Weinberg Foundation and the EAD, focused on setting up a fiduciary role, identifying additional funding sources, and developing a funding distribution process.

The KFSC began delivering food to older adults in need from the first month of its formation, with member organizations coordinating their efforts to ensure efficient distribution. Through partnerships with local farmers, food banks, and meal providers, the KFSC was able to quickly scale up its operations to meet the growing demand. By the end of the first month, the KFSC had served approximately 3300 older adults.

By the second month of its formation, the KFSC established a Steering Committee and added a Data Committee in addition to the three subgroups (food resources, funding, and access), increasing the structure and efficiency of the coalition (Figure 3).

In the third month, the KFSC began raising funds through multiple channels: support from the Weinberg Foundation, proceeds from a community telethon, and the local government. These initial funds allowed the KFSC to formalize a backbone support role managed by the Hawaiʻi Public Health Institute (HIPHI), which provided fiduciary and operational support. These funds were also distributed to the KFSC member organizations through an initial round of grants (Figure 3) to increase the capacity of food delivery and provide assistance for older adults to enroll in the Supplement Nutrition Assistance Program. During this month, while still focusing on emergency response, the Steering Committee developed a Logic Model (Figure 4) and adopted the CI framework to guide its efforts in support of the common goal of addressing food insecurity among older adults. This framework was later formally documented in September, along with clear definitions of the coalition’s roles and common agenda.

In the fourth month, additional funding associated with the Coronavirus Relief Fund was made available via local government agencies, which allowed for the second grant cycle that supported the KFSC member organizations in providing vulnerable older adults with food support and wrap-around services, including friendly phone calls, grocery shopping, nutrition education, and referral services.

By leveraging the expertise and resources of its diverse member organizations, the KFSC developed a comprehensive approach to addressing food inaccessibility among older adults, including meal delivery, food box distribution, and connecting seniors with additional support services. Prior to the KFSC’s formation, these organizations collectively served approximately 3000 older adults per week. Through coordinated efforts and additional resources, significant funding came through multiple channels: USD 3 million from the Federal CARES Act allocated via the City and County of Honolulu, plus over USD 400,000 from philanthropic organizations, individual donors, and local government. By December 2020, approximately 97% of the total funding had been utilized to meet the immediate needs of vulnerable seniors during the height of the pandemic. As a result of the KFSC’s efforts over the first 9 months of its existence (from late March to December 2020), the number of seniors served more than doubled to approximately 8300 older adults per week, with a total of 1.2 million meals provided to vulnerable seniors in Honolulu County.

### 3.2. Analysis by CI Condition

Kania and Kramer note that successful CI initiatives typically achieve the best outcomes when all five conditions are met [18]. Using the Collective Impact Assessment Rubric developed by Lynn et al. [30], this study evaluated the KFSC’s implementation of each of the five conditions, revealing varying levels of maturity across components. Details of the CI assessment rubric used and quotes supporting the findings are shown in Table 3.

#### 3.2.1. Common Agenda

The KFSC demonstrated an advanced stage of maturity in its common agenda, meeting all four Assessment Elements of this condition. A key theme that emerged through the analysis was that the urgency of addressing the food security of seniors due to the pandemic was a catalyst and aligning force that resulted in the coalition members’ willingness to work together to find solutions (Assessment Element Common Agenda (CA) 1, Table 3). This urgency, combined with the short-term limited availability of food resources from restaurants and hotels that were suddenly closed, meant that the coalition initially played a primarily coordinating role, connecting food providers with meal production and delivery resources during the first two months:


*“What is really unique about this coalition is that all the coalition members put aside individual interest and ego and think what’s best for kūpuna, lets understand where we are and where we’re going.”*
(Notes from an interview with a funder, 11/11/20.)

A powerful example of this shared vision was the response to an urgent gap in meal preparation and delivery that occurred a few months into the KFSC’s existence. In May 2020, one of the largest meal providers was forced to suddenly stop providing meals due to operational and financial issues. The remaining KFSC members quickly mobilized, leveraging resources to contact affected elder residential buildings and register seniors with replacement providers. Within days, other meal providers adjusted their distribution to ensure continued service coverage (CA2 and CA3):


*[Member H] is still awaiting direction from DOH (Department of Health), so they are not currently serving meals […] Coalition now has all of the buildings covered (that [Member H] used to deliver to) with replacement providers, with just a couple of overlaps […] Will be contacting each meal provider to confirm their list with start dates.*
(Steering Committee meeting notes, 22/05/20.)

As the initial emergency response urgency shifted to a focus on sustainability in the July/August timeframe (Figure 3), the alignment on a common agenda began to suffer. The KFSC previously lacked documentation around the common agenda and the role of the KFSC (CA4). Additionally, as the importance of the coordination role decreased as the pandemic impacts extended, the KFSC began to actively look for other ways to create value for members, including forming the Funding Committee (evolved from the funding subgroup) to identify and solicit additional funding sources:


*While this [coordination role] helped with the immediate emergency response needs, there was a need for funding support to grow and sustain the efforts of the coalition members. To meet this need, several organizations that were involved in the initial launch efforts came together to map out potential funding sources.*
(After-Action Report, 2021.)

The leadership of the KFSC recognized this gap in alignment and developed and documented a common agenda. While the main participants involved in the development process were aligned around the common goal, some of the less involved members were not as aware of the focus. The goals and action items were shared with all the member organizations through weekly general meetings, with discussion to ensure that there was not any confusion or disagreement. While the survey results indicate strong alignment—over 87% of members agreed or strongly agreed that the vision statement aligned well with the KFSC’s work (CA4)—there was very little input received. This is evidenced by the number of participants in each general meeting declining by 50% from the first three months of the KFSC formation (March to May) to the fourth quarter of 2020 (from 47 to 24 participants on average).

#### 3.2.2. Shared Measurement System

The KFSC demonstrated an advanced stage of maturity in its shared measurement system through standardized metrics and consistent data collection processes, meeting all three Assessment Elements of this condition. The shared measurement system worked well in reporting outcomes at a coalition and member level, and the data were used heavily to make evidence-based decisions. 

Early in the KFSC’s existence, members agreed that they needed to make evidence-based decisions and progress needed to be measurable. Given the emergency response nature of the initial effort, the KFSC needed to identify the scale of response required by understanding the number of seniors in potential need of food at a detailed geographic level.

The KFSC created a common set of metrics (number of seniors served, meals delivered, and food boxes provided) (Assessment Element Shared Measurement (SM) 1, Table 3) and an online data collection form that members were expected to update weekly (SM2). This common set of metrics and comparison of actual results and potential needs provided a shared measurement system.

A key theme that emerged across the documents and surveys was the power of this shared measurement system. It enabled coalition members to learn from each other’s performance and documented the KFSC’s progress as a whole (SM3). Additionally, because this success was being tracked at a detailed geographic level, the use of mapping and data visualization tools allowed organizations to quickly see their impact and have a clearer understanding of opportunities (SM2):


*The x-axis on the graph is the number of potential seniors in need in each ZIP code (10) and the y-axis is the number of seniors being served. Based on this, we can see that the ZIP codes that we are serving the best are […] and the ZIP codes where we have the most opportunity are […].*
(KFSC Mapping Summary slides through 072520. Shared in the general meeting, July 2020.)

An added benefit of this shared measurement system was that it provided a common language between meal providers and potential funders. Meal providers could identify where there was additional demand and where their resources were needed and clearly communicate incremental funding needs. Funders could optimize the impact of their funding by ensuring that there was minimal overlap and that resources were funneled to the most efficient and effective providers (SM3):


*[…] Coming into the pandemic, this was the group most committed to using data and stories for making the case for how we can work together to make a collective impact.*
(Notes from interview with a funder, 11/11/20.)

The major challenge to delivering this shared measurement system was that over half the member organizations lacked the capability or capacity to track and aggregate their data. Many organizations were non-profits with limited staff who had the capacity or skill set to collect and report data, making it hard to aggregate data weekly. To address this, the KFSC backbone created a support role to answer questions and coordinate responses, provided individualized reports to members highlighting their organization’s impact, and leveraged funds raised to incent reporting compliance (SM2):


*I’ve asked (my team) to include our development department into (the impact report to the board members and funders), as it’s a report.*
(Email communication from a member organization, 17/06/20.)

#### 3.2.3. Mutually Reinforcing Activities

The KFSC demonstrated a moderate level of maturity in mutually reinforcing activities, meeting three of four assessment elements of this condition. From launch, members worked together to deliver meals to seniors in need as quickly as possible, based on the skill sets and resources they possessed (Assessment Element Mutually Reinforcing activities (MR)1, Table 3):


*[Member C]—convener of network*



*[Member D]—connection to Kūpuna who are calling in for support*



*[Member E]—getting volunteers to help get resources to kūpuna […] Happy to work with any group that needs volunteers—can work with each group to capture the needs and then get that information out.*
(General meeting notes, 22/03/20.)

The subgroup structure of the KFSC also provided for mutually reinforcing activities. The separation of roles by the subgroups, with connections between members being required, ensured tight relationships among members (MR3). Additionally, the KFSC pushed for sharing best practices, needs, and lessons learned as a part of the general meetings. By sharing stories and successes, there was mutual recognition and development of solutions across member organizations (MR4):


*[Member G] mentioned that, because of high demand, many of the resources are quickly being used, please continue to coordinate through […] opportunities up on the site […]*



*The (web)site continues to provide updates and please share and respond to opportunities as appropriate.*
(General meeting notes, 13/04/20.)

The success of the initial emergency response effort (more than doubling meals provided in the first two weeks) and the strength of the mutually reinforcing activities allowed the coalition to quickly mobilize significant resources as large-scale funding emerged, including USD 3 million in Federal CARES Act funding through the City and County of Honolulu and an additional USD 2 million from the U.S. Administration of Community Living via the State of Hawaiʻi Executive Office on Aging, further reinforcing for members the value and impact of being a part of the KFSC.

However, the larger funding pools resulted in the KFSC implementing grant management processes in July 2020. There were now winners and losers among the members when it came to funding distribution, and members started to focus on the sustainability of their individual operations (MR2). There was also less time spent on sharing success stories or needs and less focus on resolving challenges and creating partnerships as the backbone support organization shifted its limited resources to managing funding pools and grant expectations instead:


*“The introduction of the grant and contract separated the smaller nonprofits from having a voice at the table. Felt like the larger organizations pushed out the nonprofits that are hands-on with kūpuna in the community.”*
(Learning Circle Discussion survey, 27/08/20.)

The Steering Committee recognized this issue and implemented several initiatives to increase mutually reinforcing activities. They conducted learning circle discussions, which brought together grantees to share lessons learned, strengthen common experiences, and help solve issues through the members’ expertise (MR3). Additionally, in August 2020, the KFSC launched an education and advocacy effort to increase the number of seniors receiving SNAP benefits. One member leveraged education previously developed to teach other organizations how to educate seniors on the SNAP program (MR2). Another member identified legislative efforts around SNAP expansion, and most members engaged in an advocacy effort focused on increasing access and benefits (MR3).

However, beyond the initial ramp-up period, the KFSC lacked a shared action plan that specified strategies and actions that different partners committed to implementing. The added mutually reinforcing activities were not sufficient to overcome the slow fragmentation of the broader coalition, especially since two of the primary activities only included the members who successfully competed against each other to win grants.

#### 3.2.4. Continuous Communication

The KFSC demonstrated a moderate level of maturity in continuous communication, meeting two of the four Assessment Elements of this condition. The KFSC had a significant head start since many of the member organizations and individuals had previously worked together on other projects. There was already a level of trust that allowed for more efficient communications and faster action. Additionally, the initial leadership for the KFSC included a trusted government party and a trusted community organizer, which increased the comfort level of the member organizations.

The committee structure of the KFSC also created continuous communication. The working groups’ activities were shared through the Steering Committee and the coalition meetings (Assessment Element Continuous Communication (CC)1, Table 3). Through this communication within each working group and in the general sessions, individual members became more effective, maximizing their impact and using their resources more efficiently (CC2):


*Update from] SNAP outreach and enrollment subcommittee […] looking at opportunities for volunteer network to reach out […] trying to understand how to report it out—[Individual 3] to send [Individual 4] info and an update […].*



*Data Committee Needs / Updates […] midpoint learning circle is this Thursday […]. Now have full meal data through 25 July […].*
(Steering Committee meeting notes, 20/08/20.)

This initial focus on communication weakened over time. In particular, the transition of the community organizer role occurred prior to necessary resources being in place, and the interim caretakers did not have the capacity to connect or prepare for the meetings or provide cross-sector communication or problem-solving. As a result, communication on actions, successful partnerships, and opportunities began to fade out. As dedicated leadership was put in place, the focus on continuous communication improved, with consistent meeting cadence, published agendas and minutes, and sharing of successes and best practices (CC1). Members welcomed this change, and their rating on the topic of “The meetings, and communications of the program keep you up to date on individual and collective work” increased from 3.75 out of 5 to 4.64 out of 5 between August and November.

However, the evidence for external communication structures and processes (Assessment Element CC3, Table 3) was more limited, as was evidence that these communications effectively informed and engaged the public about the initiative (Assessment Element CC4, Table 3). While internal communication mechanisms helped sustain partner coordination, the KFSC’s ability to maintain consistent and comprehensive external communication remained an area for development.

#### 3.2.5. Backbone Function

The KFSC demonstrated a moderate level of maturity in backbone support functions, meeting two of the four Assessment Elements of this condition. Initially, the AARP served as the backbone organization, with dedicated staff to perform backbone functions (Assessment Element Backbone Support (BS)1, Table 3). The effectiveness of this early leadership was noted by a Steering Committee member:


*“[Individual 4] helped hold everything together; she was the point person between the Coalition and the individual members when it was time to be in direct contact with people. It might not have worked as well as it appears that it did if we didn’t have a strong lead like […]”.*
(Notes from an interview with a Steering Committee member, 11/11/20.)

However, the KFSC faced challenges in establishing a well-functioning leadership structure for governance and decision-making (Assessment Element BS2, Table 3). While most member organizations felt that the natural leadership for the KFSC should come from the government agency that initiated the effort, the agency strongly believed in the need for community leadership rather than a top-down approach. While this worked successfully early on, mainly due to the strong participation of the community organizer, the AARP, in consistently moving the effort forward, it started to falter with the transition to the interim organizer, which saw its role as limited to being a support organization and looked to the Steering Committee to determine strategic direction and focus:


*“We haven’t figured out the governance, decision-making process. It’s respectful of those good relationships and trust to formalize the process. We have committees but not yet a clear process as to grants, etc.”*
(Notes from an interview with a funder, 11/11/20.)

The backbone infrastructure coordinated and supported core initiative activities (Assessment Element BS3, Table 3), including consolidating information about food resources, delivery services, and volunteer services through established channels. When the HIPHI took over backbone support in June, the transition initially presented challenges. The interim coordinator’s brief two-month tenure was marked by organizational and support difficulties that temporarily affected KFSC operations. However, the subsequent appointment of a new coordinator helped stabilize operations and strengthen coordination efforts. The backbone staff demonstrated appropriate skills and credibility (Assessment Element BS4, Table 3) in supporting partners’ work, particularly in understanding the importance of comprehensive service delivery:


*[Individual 5 shared] the importance of wrap-around services when delivering food. “Being able to check in with folks—it really is saving lives. Reminds us of how important this work is.”*
(General meeting notes, 28/09/20.)

These observations about backbone support and leadership transitions would prove significant in understanding the KFSC’s overall development, as discussed further in the discussion section.

#### 3.2.6. Additional Themes

Beyond the five CI conditions, an additional theme emerged from the four data sources, which was the importance of sustainable funding. Early on, much of the KFSC’s work was supported by volunteers, with seed funding from private foundations and a community telethon (March through May, Figure 3). The influx of Federal CARES Act funding in June allowed the KFSC to scale its operations quickly and secure backbone support to manage the increased resources and coordination efforts.

The availability of these funds was crucial in enabling the KFSC to respond effectively to the sudden increase in food insecurity among older adults caused by the pandemic. The additional resources allowed member organizations to expand their services, hire additional staff, and purchase necessary equipment and supplies to meet the growing demand for food assistance. The backbone support provided by the HIPHI was also essential in managing the increased funding and ensuring that resources were distributed efficiently and equitably among member organizations.

However, as this funding diminished, the need for long-term funding became urgent. While additional funding rounds were secured, they were inconsistent, leaving many members without the resources to sustain previous levels of service: 


*If we are able to find additional funding, we would keep feeding these kupuna communities for as long as possible. As of right now, we will be ending once the […] program concludes.*
Learning Circle Discussion survey, 11/6/20)


*Our whole […] project was possible because of this grant which currently serves over 3000 people each week. This grant has allowed us to partner with social service agencies and provide meals and services to vulnerable families and kūpuna all over the island. At the end of […] funding, this program will be done since we do not have other funding to sustain it.*
(Learning Circle Discussion survey, 22/06/20.)

The challenges of funding sustainability and their impact on service delivery capacity emerged as critical factors affecting coalition maturity, as analyzed in detail in the Discussion section.

## 4. Discussion

The goal of this study was to evaluate the operation of the Kūpuna Food Security Coalition using the CI framework to assess the replicability of the approach in addressing increased food access among older adults during the COVID-19 crisis. The findings provide insights for other public/non-profit partnerships aiming to replicate the KFSC’s efforts in emergency response situations.

Although not originally designed as a CI initiative, the KFSC incorporated the five CI components from very early in its formation. There was a clear common agenda due to the urgency of the situation, a shared measurement system that delivered alignment and success metrics, mutually reinforcing activities that were structurally embedded, continuous communication delivered through regular meetings and learning circle discussions and a dedicated backbone support staff that increased the effectiveness and efficiency of the KFSC’s efforts. By the end of nine months, two of the five CI conditions demonstrated advanced levels of maturity, and the other three demonstrated moderate levels of maturity.

The KFSC’s approach demonstrated success as a short-term emergency response, effectively managing funds and distributing food to address the immediate needs of vulnerable older adults. The KFSC served approximately 1.2 million meals to 8300 seniors in Honolulu County over its first nine months, showcasing its ability to quickly mobilize resources and coordinate efforts among diverse organizations.

Despite this success in emergency response, two significant gaps emerged. The first gap was the lack of consistent leadership, which is crucial for coalition success [41]. Zakocs and Edwards investigated 26 existing studies to learn which factors indicate successful coalition-building and its effectiveness. The researchers found that leadership is one of the key success components [12]. Hanleybrown et al. emphasized the importance of an influential champion who can bring together and engage cross-sector leaders [42]. In the early phase (the first two months) of the KFSC, the leadership of the EAD and AARP’s role as a proactive facilitator worked well to mobilize and guide the KFSC. The existing literature highlights that this facilitating leadership style is recommended for coalitions as they constantly learn, share, and encourage together [43,44]. However, when the HIPHI took over as the backbone organization, an initial two-month period with a temporary coordinator who lacked the necessary organizational and leadership skills led to operational challenges. While a subsequent coordinator helped stabilize operations, this transition period highlighted how leadership changes can significantly impact coalition momentum and effectiveness.

The second critical gap identified was the lack of sustainable funding mechanisms. Many innovative public policies and programs at the community level, initially supported by public funding, are not sustainable when the funding is terminated [21,45,46,47]. Hanleybrown et al. stressed the need for adequate financial resources lasting at least two to three years, typically from an anchor funder engaged from the beginning [42]. The KFSC’s early success was due to significant emergency response funding from the federal government, private foundations, and public donations, which supported the backbone structure and brought member organizations together. Lynn et al. found that backbone support is the most crucial component to implement and sustain initiatives [35]. This backbone role is not possible without consistent funding.

Despite these challenges, the approach followed by the KFSC’s experience offers valuable lessons for future emergency response efforts. The CI framework provided a useful guide for structuring the KFSC’s work, with key components particularly standing out, including (1) establishing a common agenda and documenting early and reinforcing it often, (2) developing a shared measurement system, which is a powerful tool for alignment and communication, and (3) mutually reinforcing activities and internally managed grant processes, which may come in conflict with each other. Additionally, the KFSC’s ability to quickly mobilize resources and coordinate efforts among diverse organizations was a key strength, highlighting the importance of pre-existing relationships and networks in emergency response.

The experience of the KFSC experience also underscores the critical importance of having strong backbone support, an active leadership role as a facilitator, and securing consistent funding for the sustainability needed to address longer-term solutions for food access.

While the CI framework provided valuable structure, our findings suggest several important considerations beyond the framework’s scope. First, the framework does not fully account for coalition evolution over time, as evidenced by the KFSC’s natural progression from emergency response to broader community impact. The KFSC’s varying levels of maturity across different CI conditions raise questions about whether all pillars carry equal weight in emergency response situations. Additionally, factors such as relationship building, trust, and absence of ego among members emerged as critical elements that enabled effective collaboration but are not explicitly addressed in the CI framework.

The KFSC’s success in emergency response was significantly enhanced by its members’ deep understanding of the local context and cultural competency. Many KFSC members were well-established organizations with extensive experience serving older adults in Honolulu County, bringing not only service capacity but also crucial knowledge of cultural preferences, trust relationships, and community needs. This integration of cultural understanding and community voice, particularly in Hawaiʻi’s diverse cultural context, enabled the KFSC to effectively tailor its approach and ensure services reached those most in need, demonstrating the value of working “with” rather than “for” communities.

While this study provides valuable insights into the KFSC’s implementation of the CI framework, its focus is on a single community setting, which limits the generalizability of its findings. Future research examining multiple sites implementing CI approaches to address older adult food insecurity would strengthen the evidence base.

While not part of this study, another indicator of KFSC’s success was its continuation beyond its initial emergency response efforts. The KFSC remained in operation through 2021, with many members then joining a similarly structured community coalition, the Kūpuna Vaccination Outreach Group (KVOG), to manage COVID-19 vaccinations across the state. The work of this spin-off group resulted in the vaccination of 94% of the state’s older adults by 30 July 2021 [48]. In February 2022, the KFSC and KVOG came together to form the Kūpuna Collective, bringing together a collaborative network of partners to elevate critical issues, mobilize community assets, and drive innovative solutions to support and empower kūpuna [49]. This evolution into the Kūpuna Collective demonstrates how cross-sector partnerships can effectively address not just food security but broader social determinants of health and drive innovative solutions that support and empower older adults in the long term.

As communities continue to grapple with the ongoing impacts of the COVID-19 pandemic and other emergencies, the need for effective, coordinated responses to address food security among vulnerable populations remains urgent. By applying the lessons learned from the KFSC and adapting the CI framework to their specific contexts, public/non-profit partnerships can build more resilient, equitable, and sustainable food systems that prioritize the needs of older adults and other at-risk groups.

## 5. Conclusions

The CI framework provided valuable structure for the KFSC’s work, yet this study suggests several critical adaptations needed for emergency response applications. First, while the framework emphasizes backbone support, our findings indicate the need to specifically address leadership continuity and clearly defined roles during leadership transitions. Second, whereas the current framework assumes sustained funding, emergency response initiatives require guidance on securing and managing diverse, time-limited funding streams. Third, there is a need for continuous adaptation and learning as coalitions evolve, rather than a rigid or formulaic approach to the implementation of the CI framework.

Additionally, our analysis reveals the importance of elements not explicitly addressed in the CI framework but crucial for emergency response success: (1) the critical role of relationship building and trust alongside structural elements; (2) the necessity of dedicated leadership capacity rather than adding coalition responsibilities to existing workloads; and (3) the value of strong community voice and engagement, particularly in diverse cultural contexts. These factors proved essential for rapid mobilization and effective service delivery during crisis response.

The KFSC demonstrated how emergency response coalitions, when built on a foundation of trust, shared values, and cultural understanding, can effectively address immediate needs while building capacity for broader community impact. This emergency response model adopted by the KFSC later evolved into the broader Kūpuna Collective [44], demonstrating how well-structured crisis response initiatives can create foundations for sustainable community partnerships. This case study provides valuable guidance for public health organizations, aging service providers, and emergency response planners seeking to establish effective public/non-profit partnerships during crises, with the potential for longer-term community impact. The approach can be particularly beneficial for communities needing to rapidly mobilize resources and coordinate multi-stakeholder efforts in emergency situations while building capacity for future collaborations.

## Figures and Tables

**Figure 1 ijerph-22-00163-f001:**
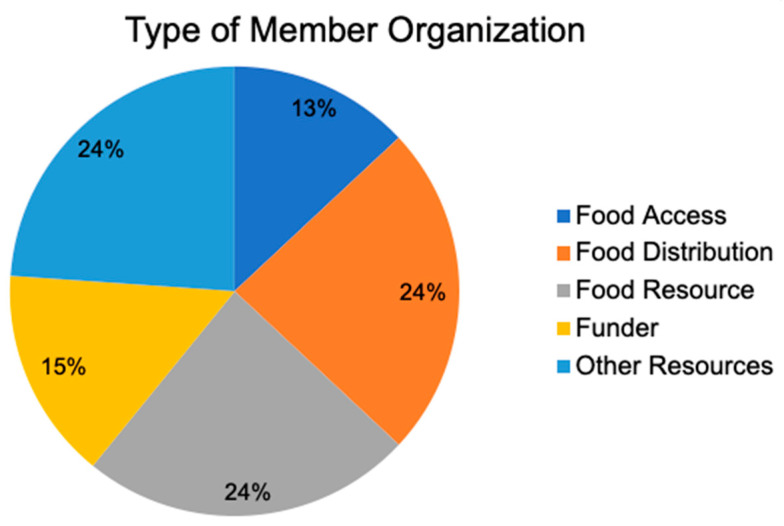
KFSC type of member organization.

**Figure 2 ijerph-22-00163-f002:**
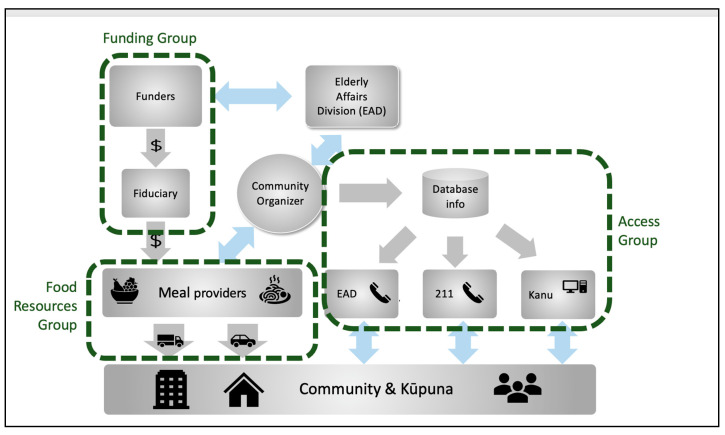
Initial KFSC operating model.

**Figure 3 ijerph-22-00163-f003:**
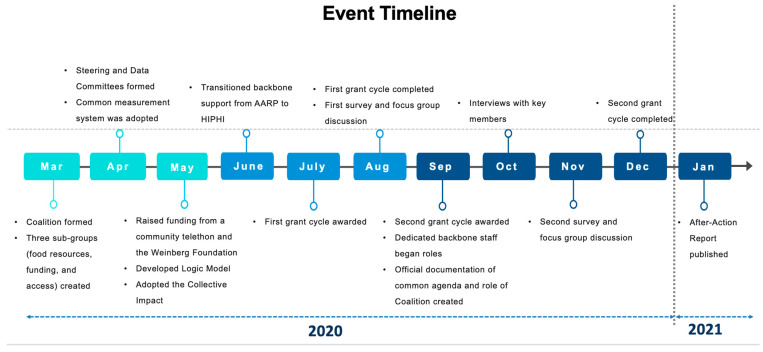
KFSC timeline.

**Figure 4 ijerph-22-00163-f004:**
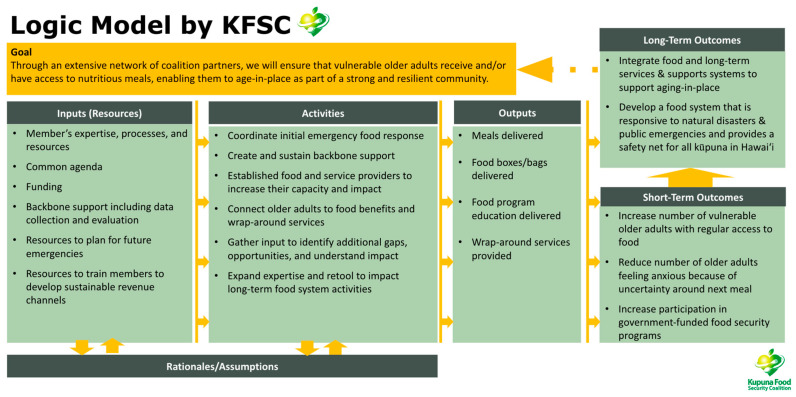
KFSC Logic Model.

**Table 1 ijerph-22-00163-t001:** Data source questions and analysis method (excluding management documents).

	Questions	Analysis Method
Interviews	What did you learn from your role? What worked well and didn’t work well? Any opportunities that we’re not taking full advantage of? What would you recommend that we do differently next time? What should we have asked you that we didn’t?	Deductive thematic analysis was applied to categorize transcripts under the CI framework.
Focus group discussions	Success stories/positive impact of efforts. What successes are you seeing? What success stories could you share? What are you most proud of with your current initiative? Challenges/pain points—shared learnings. What challenges are you facing with being able to achieve your objectives? Operational, budget, staffing, etc. How are you identifying vulnerable elders? How is the process of identifying vulnerable seniors working? What unexpected needs are you seeing? Course corrections/planned adjustments. What lessons have you learned in recent weeks, and what changes will you potentially make as a result?	Deductive thematic analysis was applied to categorize transcripts under the CI framework.
Surveys	(Likert scale) Vision Statement—The coalition is aligned around the vision statement of “Vulnerable kūpuna on `Oahu are food secure and connected to needed wrap-around services to support aging-in-place; a food system is created to build a strong and resilient community.” (Open-ended) Does this reflect your understanding of the role of the coalition? If your answer is no, how would you revise this statement? (Likert scale) Continuous Communication—The meetings and communications of the coalition keep you up to date on individual and collective work. (Likert scale) Partnership Collaboration—The coalition is resulting in partners working together to achieve more than they would working separately. (Likert scale) Role of HIPHI (KFSC’s backbone organization)—The structure and logistics maintained by HIPHI help ensure the success of the coalition. (Likert scale) Coalition model allows for a diverse set of voices from appropriate sectors. (Open-ended) If you disagree, what would you change, or who else should be a part of the coalition? (Likert scale) The current set of data measures does a good job of tracking our progress and outcomes. (Open-ended) Is there additional data we should be looking at to examine progress and outcomes? (Open-ended) What changes would you recommend making the coalition more successful/impactful in achieving the shared vision?	Analyzed for frequency distribution using Microsoft Excel, while open-ended responses were thematically coded.

**Table 2 ijerph-22-00163-t002:** KFSC’s management documents.

Source of Evidence		Number of Documents	Analysis Strategy
Categories	Description		
Operational documents	Member listing (included all members’ names, organizations, and contact info)	*n* = 1	Thematic analysis to identify roles, key dates, decisions, resource allocations, participation, engagement, successes, opportunities, and lessons learned and categorize relative to CI framework
	Strategy documents	*n =* 2 (KFSC overviews)	
	Meeting notes (included topics discussed, decisions made, timelines, resources, etc.)	*n* = 60 (19 general meetings, 31 Steering Committee meetings, and 10 Funding Committee meetings)	
	Email communications (unstructured data)	*n* = 806 (To and from the author between Mar. 2020 and Feb. 2021)	
Grant documents	Grant applications from member organizations (included member organizations’ mission, goals, targeted population served, and types of services they provide)	*n* = 65 (Multiple documents per application in two grant cycles)	Thematic analysis to identify service to seniors and extract info on mission, goals, etc., to categorize relative to CI framework
	End-of-grant reports from member organizations (included what work was undertaken, how many served, and where they served)	*n* = 35 (One final report with other supporting documents from each grant recipient in two grant cycles)	
External presentations and reports	External reports/presentations to funders and government agencies (included an after-action report and presentations used for funding solicitation purposes, etc.)	*n* = 8	Thematic analysis to identify vision and mission, additional info on activities, participants, resource use, impact, and backbone support relative to CI framework
	MOUs and agreements with backbone organization	*n* = 1	
Data tracking and reports	Weekly data and other reports from member organizations (included data on services provided and seniors served)	*n* = 176 (Most members submitted a weekly report as well as other summary reports)	Content analysis (36) to identify services to seniors and timelines for members to submit their data

**Table 3 ijerph-22-00163-t003:** Collective Impact Assessment Rubric [30].

Condition	Collective Impact Assessment Elements
Common Agenda	CA1: Identifiable overarching goal and vision for initiative within clearly defined, bounded/actionable problem space CA2: Partners have a common understanding of problems CA3: Partners have a clearly articulated approach/set of high-level strategies to solve problems CA4: Partners have a high level of buy-in to a shared vision for change, agreed-upon goals, and approaches
Shared Measurement System	SM1: Agreed-upon common indicator(s) established to consistently track progress across time SM2: Functional approach and system to collect, store, analyze, and report valid and reliable data SM3: Output/results of the shared measurement system are actionable for data use (timely, meaningful, relevant, sensitive to change, targeted to goal, etc.)
Mutually Reinforcing Activities	MR1: Collective action plan specifying strategies and actions different partners commit to implement MR2: Partners implement strategies to advance the shared action plan MR3: Working groups/collaborative structures established to coordinate activities aligned with the action plan MR4: Partners hold each other accountable for implementing activities as planned
Continuous Communication	CC1: Structures and processes in place to inform, engage, and seek feedback from internal partners CC2: Internal communications support the effective functioning of initiative work CC3: Structures and processes in place to inform and engage the public/community about initiatives CC4: External communications inform and engage the public about initiatives, facilitate knowledge and understanding, increase buy-in to initiatives, and provide opportunities for feedback and input
Backbone Support Organization	BS1: One or more orgs with committed staff designated to perform backbone functions BS2: Well-functioning leadership structure established, responsible for governance and decision-making BS3: Backbone infrastructure coordinates and supports core initiative activities BS4: Backbone staff have appropriate skills and credibility to perform backbone functions

## Data Availability

The data presented in this study are available upon request from the corresponding author.

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
