# Peer review of "Assessing a Community Partnership Addressing Food Security Among Older Adults During COVID-19"

_ijerph, 2025, doi:10.3390/ijerph22020163_

Round 1
Reviewer 1 Report
Comments and Suggestions for Authors
1. First of all, the researchers have not fully defined how they explored food insecurity among older adults and identified those experiencing food insecurity. This lack of clarity raises questions about the methods used for data collection and analysis, which are crucial for ensuring the validity and reliability of the findings. Understanding the specific criteria and tools utilized to assess food insecurity is essential, as this could directly impact the identification of at-risk individuals. For instance, were standardized questionnaires employed, or were qualitative interviews conducted? Additionally, the demographics of the sample population play a critical role in how representative the findings are. A detailed description of the research methodology would enable greater insight into the scope of the study and its applicability to broader populations.
2. A section detailing the significant obstacles encountered in maintaining stable leadership and securing reliable funding should be added to the manuscript. These challenges are critical, as they pose a threat to the coalition’s long-term sustainability and effectiveness. Without consistent leadership and adequate financial resources, the coalition may struggle to adapt to emerging circumstances or effectively expand its initiatives. Addressing these issues in the manuscript would provide valuable context for understanding the coalition's operational challenges and could highlight the importance of strategic planning in overcoming such barriers.
3. There is a lack of coherence throughout the manuscript, as the abbreviations are reiterated without acknowledgment of their initial definitions.
4. Including a table that outlines the program strategies and action plan would add significant value to the manuscript. Such a table can provide readers with a clear, concise overview of the key components of the program, making it easier to understand the specific strategies being implemented and their corresponding actions. This visual representation would not only enhance the organization of the information but also facilitate quick reference for readers, allowing them to grasp the program's objectives and practical steps at a glance.
5. As the current program tries to shift from an emergency response approach to a long-term food security strategy, it is vital to implement sustainability mechanisms that extend beyond initial funding sources. This involves promoting innovation within partnerships and developing pathways for continued community support.
Author Response
Comment 1: First of all, the researchers have not fully defined how they explored food insecurity among older adults and identified those experiencing food insecurity. This lack of clarity raises questions about the methods used for data collection and analysis, which are crucial for ensuring the validity and reliability of the findings. Understanding the specific criteria and tools utilized to assess food insecurity is essential, as this could directly impact the identification of at-risk individuals. For instance, were standardized questionnaires employed, or were qualitative interviews conducted? Additionally, the demographics of the sample population play a critical role in how representative the findings are. A detailed description of the research methodology would enable greater insight into the scope of the study and its applicability to broader populations.
Respose 1:
Thank you for this comment. This highlighted that the text created some confusion as to the focus of the study. As such, we updated throughout to clarify that the primary focus of this manuscript is not to explore or identify food insecurity among older individuals, but rather to evaluate the effectiveness of a community coalition in providing food for older adults as an emergency response to increased food insecurity during the COVID-19 pandemic. Our study specifically examines how well the Kūpuna Food Security Coalition operationalized to coordinate 40+ organizations in delivering food services to vulnerable older adults by utilizing the Collective Impact framework. The coalition worked with existing aging services networks and community organizations that had already identified older adults needing food assistance. As such, this study focuses on the effectiveness of the coalition structure in delivering food to older adults rather than the effectiveness of that food in reducing food insecurity.
Comment 2: A section detailing the significant obstacles encountered in maintaining stable leadership and securing reliable funding should be added to the manuscript. These challenges are critical, as they pose a threat to the coalition’s long-term sustainability and effectiveness. Without consistent leadership and adequate financial resources, the coalition may struggle to adapt to emerging circumstances or effectively expand its initiatives. Addressing these issues in the manuscript would provide valuable context for understanding the coalition's operational challenges and could highlight the importance of strategic planning in overcoming such barriers.
Response 2:
Thank you for this suggestion as it highlights the importance of these findings. To reinforce these points, we have made the following changes:
- Regarding leadership, we have included more details in lines 429 to 465 on how the leadership status impacted the operations of the KFSC in Section 3.2.5 (Backbone Function).
- We updated the Discussion Section on how the leadership status was changed from challenging to strong in lines 525 to 530, reflecting Section 3.2.5 (Backbone Function).
We completely agree that both consistent leadership and adequate financial resources are critical to the coalition's long-term sustainability and effectiveness. However, since this study focuses on the coalition's first nine months of operation, we limited our discussion to the impacts observed during this period while noting that these factors may have greater implications for the future.
Comment 3: There is a lack of coherence throughout the manuscript, as the abbreviations are reiterated without acknowledgment of their initial definitions.
Response 3:
Thank you for this feedback. We have conducted a thorough review of abbreviation usage throughout the manuscript to ensure each term is properly defined at first use and used consistently thereafter. The key abbreviations in our manuscript include:
- KFSC (Kūpuna Food Security Coalition)
- CI (Collective Impact)
- EAD (Elderly Affairs Division)
- HIPHI (Hawaiʻi Public Health Institute)
- AARP (formerly American Association of Retired Persons)
We’ve ensured that:
- Each abbreviation is spelled out in full at first use
- The abbreviation follows in parentheses
- Only the abbreviation is used in subsequent references (except one reuse of Kūpuna Food Security Coalition in the conclusion which we felt was appropriate)
- Consistency is maintained throughout all sections
- Instead of using Kūpuna Food Security Coalition (Coalition), we used KFSC throughout the manuscript for clarity.
Comment 4: Including a table that outlines the program strategies and action plan would add significant value to the manuscript. Such a table can provide readers with a clear, concise overview of the key components of the program, making it easier to understand the specific strategies being implemented and their corresponding actions. This visual representation would not only enhance the organization of the information but also facilitate quick reference for readers, allowing them to grasp the program's objectives and practical steps at a glance.
Response 4: Thank you for this feedback. Please see the updated (line 212). We have included the Logic Model developed for the Kūpuna Food Security Coalition as a new figure in the document.
Comment 5. As the current program tries to shift from an emergency response approach to a long-term food security strategy, it is vital to implement sustainability mechanisms that extend beyond initial funding sources. This involves promoting innovation within partnerships and developing pathways for continued community support.
Response 5: Thank you for this thoughtful suggestion about sustainability mechanisms. We agree that long-term sustainability is an important consideration for public/non-profit partnerships. However, since the primary focus of this manuscript is to evaluate the KFSC's effectiveness as an emergency response initiative during its first nine months of operation (March-December 2020), we intentionally concentrated on this critical period to provide detailed insights into how public/non-profit partnerships can effectively mobilize and coordinate resources to address the intended outcomes during crisis situations. While we briefly note the KFSC's evolution into the Kūpuna Collective to demonstrate the potential for emergency response initiatives to spark longer-term change, an in-depth analysis of sustainability mechanisms would extend beyond the scope of this particular evaluation. The sustainability aspects of the KFSC's transition could be valuable material for future research and we are discussing this opportunity.
Reviewer 2 Report
Comments and Suggestions for Authors
Abstract
The abstract section need to be improved by mentioning some key features of this study. Further more also add the future impact of current study.
Introduction
The introduction should be updated with recent references relevant to current study.
Results and Discussion
Please use appropriate figures to represent data where applicable.
Conclusion
The conclusion should be updated by adding the future impact of current study and potential beneficiary of the current study.
Comments on the Quality of English Language
Minor grammatical errors need to be addressed to improve the quality of the manuscript.
Author Response
Comment 1: Abstract - The abstract section need to be improved by mentioning some key features of this study. Further more also add the future impact of current study.
Response 1: Thank you for this suggestion. We have revised the abstract to better emphasize key features by adding:
- Specific details about KFSC's organizational structure and operation
- Clear metrics of success (e.g., coordination of 40+ organizations, service to 8,300 seniors)
- Impact on emergency response capabilities
Comment 2: Introduction - The introduction should be updated with recent references relevant to current study.
Response 2: Thank you for this suggestion. We did a quick update to our literature review and added a couple of additional references.
Comment 3: Results and Discussion - Please use appropriate figures to represent data where applicable.
Response 3: Thank you for this suggestion about additional figures. We chose to add one additional figure to the document (Figure 3: KFSC Logic Model). However, we chose not to add any further figures representing data since we felt that these would further extend the article without adding significant value or clarity.
Comment 4: Conclusion - The conclusion should be updated by adding the future impact of current study and potential beneficiary of the current study.
Response 4: Thank you for this suggestion. While our conclusion discusses implications for public/non-profit partnerships, we agree that it could better highlight future impacts and potential beneficiaries. In the Conclusion section, we’ve integrated comments on future impacts, including the application of the CI framework in emergency response situations, the development of rapid-response coalition models, and the enhancement of community emergency preparedness, while maintaining the conclusion's focus on the evaluation of the KFSC's emergency response efforts. We also added a brief sentence on potential beneficiaries.
Reviewer 3 Report
Comments and Suggestions for Authors
The manuscript clearly describes the purpose, methodology, and findings of the study. It provides a detailed account of the Kūpuna Food Security Coalition (KFSC) and its achievements. The use of the Collective Impact (CI) framework to assess coalition functioning is well-structured and aligns with the study's goal of evaluating effectiveness. The recommendations are practical and grounded in the findings, addressing short-term and long-term considerations.
While the text mentions challenges such as leadership continuity and funding, it does not delve deeply into why these challenges occurred or how they were addressed. For example: What specific gaps in leadership continuity arose? How were funding shortfalls mitigated during the initiative?
Although the study uses diverse data sources (interviews, focus groups, surveys, documents), the methodology section lacks detail on: Sampling methods and demographics of participants. How the CI framework was operationalized to evaluate "maturity" levels. How findings from different data sources were triangulated.
The paper critiques and recommends adaptations to the CI framework but does not provide sufficient detail on how KFSC specifically modified or implemented the framework during its operation. Case-specific insights could strengthen the discussion.
While the transition from emergency response to the broader Kūpuna Collective is highlighted, the text does not provide sufficient detail on: How this transition was managed. What systems were put in place to ensure long-term sustainability.
Although the study focuses on older adult food insecurity, the discussion could more explicitly address this demographic’s unique needs, challenges, or preferences and how KFSC tailored its approach.
Some recommendations, such as those related to leadership and funding, overlap. Consolidating them could make the text more concise and impactful.
Suggestions:
1. Include a more detailed discussion of the barriers faced by KFSC and how these were addressed.
2. Provide Case-Specific Adaptations: Highlight concrete examples of how KFSC adapted the CI framework to its local context and emergency.
3. Emphasize Sustainability Efforts: Elaborate on how the coalition transitioned to a sustainable model and the key steps taken to integrate with the broader Kūpuna Collective.
4. Focus on Older Adults: Deepen the discussion on how the coalition's work specifically addressed the needs of older adults and how these insights could generalize to similar initiatives.
Author Response
The manuscript clearly describes the purpose, methodology, and findings of the study. It provides a detailed account of the Kūpuna Food Security Coalition (KFSC) and its achievements. The use of the Collective Impact (CI) framework to assess coalition functioning is well-structured and aligns with the study's goal of evaluating effectiveness. The recommendations are practical and grounded in the findings, addressing short-term and long-term considerations.
While the text mentions challenges such as leadership continuity and funding, it does not delve deeply into why these challenges occurred or how they were addressed. For example: What specific gaps in leadership continuity arose? How were funding shortfalls mitigated during the initiative?
Thank you for this observation. We have enhanced both the results and discussion sections to provide more insights into these challenges, particularly in Section 3.2.5. We have also strengthened the connection between our Results and Discussion sections to better explain how these challenges impacted coalition operations.
Although the study uses diverse data sources (interviews, focus groups, surveys, documents), the methodology section lacks detail on: Sampling methods and demographics of participants. How the CI framework was operationalized to evaluate "maturity" levels. How findings from different data sources were triangulated.
Thank you for this observation. Your question highlighted that we hadn’t clearly identified that the interviews, focus groups, and survey had been previously conducted (by the KFSC backbone support organization) and that our study analyzed the transcripts and results of these documents. We updated throughout the document to reflect this. Additionally, regarding the assessment of the maturity levels of the CI framework, we have highlighted the assessment rubric used (in Table 3) and that the condition was considered to be at an “advanced stage of maturity” if it met all the "Assessment Elements”. If not, we evaluated the condition as having a “moderate level of maturity” since the other conditions met two to three of the assessment elements.
The paper critiques and recommends adaptations to the CI framework but does not provide sufficient detail on how KFSC specifically modified or implemented the framework during its operation. Case-specific insights could strengthen the discussion.
Thank you for the insight. We added additional detail throughout the results section to address this comment. We had originally removed much of this content to reduce the length of the article but agree that the case-specific insights strengthen the paper and provide additional insights for other organizations to leverage the learnings.
While the transition from emergency response to the broader Kūpuna Collective is highlighted, the text does not provide sufficient detail on: How this transition was managed. What systems were put in place to ensure long-term sustainability.
Thank you for this thoughtful suggestion about sustainability mechanisms. We agree that long-term sustainability is an important consideration for public/non-profit partnerships. However, since the primary focus of this manuscript is to evaluate the KFSC's effectiveness as an emergency response initiative during its first nine months of operation (March-December 2020), we intentionally concentrated on this critical period to provide detailed insights into how public/non-profit partnerships can effectively mobilize and coordinate resources to address the intended outcomes during crisis situations. While we briefly note the KFSC's evolution into the Kūpuna Collective to demonstrate the potential for emergency response initiatives to spark longer-term change, an in-depth analysis of sustainability mechanisms would extend beyond the scope of this particular evaluation. The sustainability aspects of the KFSC's transition could be valuable material for future research and we are discussing this opportunity.
Although the study focuses on older adult food insecurity, the discussion could more explicitly address this demographic’s unique needs, challenges, or preferences and how KFSC tailored its approach.
Thank you for this comment. This highlighted that the text created some confusion as to the focus of the study. As such, we updated throughout to clarify that the primary focus of this manuscript is not to explore or identify food insecurity among older individuals, but rather to evaluate the effectiveness of a community coalition in providing food for older adults as an emergency response to increased food insecurity during the COVID-19 pandemic. Our study specifically examines how well the Kūpuna Food Security Coalition operationalized to coordinate 40+ organizations in delivering food services to vulnerable older adults by utilizing the Collective Impact framework. The coalition worked with existing aging services networks and community organizations that had already identified older adults needing food assistance. As such, this study focuses on the effectiveness of the coalition structure in delivering food to older adults rather than the effectiveness of that food in reducing food insecurity.
Some recommendations, such as those related to leadership and funding, overlap. Consolidating them could make the text more concise and impactful.
Suggestions:
Thank you – addressed in the comments above
- Include a more detailed discussion of the barriers faced by KFSC and how these were addressed.
- Provide Case-Specific Adaptations: Highlight concrete examples of how KFSC adapted the CI framework to its local context and emergency.
- Emphasize Sustainability Efforts: Elaborate on how the coalition transitioned to a sustainable model and the key steps taken to integrate with the broader Kūpuna Collective.
- Focus on Older Adults: Deepen the discussion on how the coalition's work specifically addressed the needs of older adults and how these insights could generalize to similar initiatives.
Reviewer 4 Report
Comments and Suggestions for Authors
The manuscript is good, but in order to be published, it requires some improvements, which may also increase its chances of impacting the specialized literature and being cited.
Abstract
Comment 1. The abstract provides a clear summary of the study, but the clarity and conciseness of the sentences could be improved. I suggest reducing some repetitions.
Concrete examples of the results should be integrated for better relevance.
Introduction
Comment 2. The introduction presents well the general context regarding food insecurity among older people and correctly introduces the importance of theoretical frameworks, such as Collective Impact (CI).
I recommend a clearer explanation of the gaps in previous research related to the approach to food insecurity in the context of crisis.
Comment 3. In the second paragraph, the term “food in-security” needs to be corrected.
Materials and Methods
Comment 4 .The methods used are well described, including the case study approach, data triangulation and deductive thematic analysis, but the explanations regarding “data triangulation” and validation could be detailed, as they are mentioned in a summary manner.
Comment 5. Ensure that the methods applied are consistent with the study objectives, and that the limitations of the approach (e.g., participant selection, bias) are acknowledged.
Results
Comment 6. The results are presented clearly and in detail, with concrete data on the . of meals delivered and partnerships coordinated, but the graph and analysis need to be better connected to the CI framework objectives.
Comment 7. Concrete examples of the challenges encountered in implementing the action plan could be added.
Comment 8. I suggest a revision of the sub-sections for clarity, as some details are repeated.
Discussions
Comment 9. The discussions are well structured, highlighting both successes and challenges.I recommend including a comparison with other similar initiatives in other countries, to emphasize the global relevance.
Comment 10. I note that the proposed solutions for leadership and financing are solid, but their practical implications need to be explained in more detail.
Conclusions
Comment 11. The conclusions summarize the results well, but I suggest including more concrete recommendations for public policies and future initiatives. To clarify how the CI framework can be applied in other contexts or for other vulnerable groups.
General comments
Comment 12. Linguistic review - I note that the term “Food In-security” is used, although the correct form is “Food Insecurity”. This is an error that requires correction, as the correct term is one recognized and used in the scientific literature and international policies.
Comment 13. Clarity and conciseness - Some paragraphs are redundant and could be simplified to improve the flow of the text.
Addition of study limitations - A section dedicated to limitations would strengthen scientific rigor.
Author Response
Abstract
Comment 1. The abstract provides a clear summary of the study, but the clarity and conciseness of the sentences could be improved. I suggest reducing some repetitions.
Concrete examples of the results should be integrated for better relevance.
Thank you for this suggestion. We have revised the abstract to better emphasize key features by adding:
- Specific details about KFSC's organizational structure and operation
- Clear metrics of success (e.g., coordination of 40+ organizations, service to 8,300 seniors)
- Impact on emergency response capabilities
Introduction
Comment 2. The introduction presents well the general context regarding food insecurity among older people and correctly introduces the importance of theoretical frameworks, such as Collective Impact (CI).
I recommend a clearer explanation of the gaps in previous research related to the approach to food insecurity in the context of crisis.
Comment 3. In the second paragraph, the term “food in-security” needs to be corrected.
Thank you for this observation. We believe this was a formatting issue and have ensured consistency throughout the document.
Materials and Methods
Comment 4 .The methods used are well described, including the case study approach, data triangulation and deductive thematic analysis, but the explanations regarding “data triangulation” and validation could be detailed, as they are mentioned in a summary manner.
Thank you for this observation. We have updated the methods section to further clarify the approach.
Comment 5. Ensure that the methods applied are consistent with the study objectives, and that the limitations of the approach (e.g., participant selection, bias) are acknowledged.
Thank you for this observation. Your question highlighted that we hadn’t clearly identified that the interviews, focus groups, and survey had been previously conducted (by the KFSC backbone support organization) and that our study analyzed the transcripts and results of these documents. We updated throughout the document to reflect this. Additionally, regarding the assessment of the maturity levels of the CI framework, we have highlighted the assessment rubric used (in Table 3) and that the condition was considered to be at an “advanced stage of maturity” if it met all the "Assessment Elements”. If not, we evaluated the condition as having a “moderate level of maturity” since the other conditions met two to three of the assessment elements.
Results
Comment 6. The results are presented clearly and in detail, with concrete data on the . of meals delivered and partnerships coordinated, but the graph and analysis need to be better connected to the CI framework objectives.
Comment 7. Concrete examples of the challenges encountered in implementing the action plan could be added.
Comment 8. I suggest a revision of the sub-sections for clarity, as some details are repeated.
Thank you for these insights. We added additional detail throughout the results section to address these comments and worked to minimize repetition (though found that some issues had impacts across CI conditions). We had originally removed much of this content to reduce the length of the article but agree that the case-specific insights strengthen the paper and provide additional insights for other organizations to leverage the learnings.
Discussions
Comment 9. The discussions are well structured, highlighting both successes and challenges.I recommend including a comparison with other similar initiatives in other countries, to emphasize the global relevance.
Comment 10. I note that the proposed solutions for leadership and financing are solid, but their practical implications need to be explained in more detail.
Thank you for these insights. We added some additional detail throughout the discussion section to address these comments while attempting to not overly increase the length of the article.
Conclusions
Comment 11. The conclusions summarize the results well, but I suggest including more concrete recommendations for public policies and future initiatives. To clarify how the CI framework can be applied in other contexts or for other vulnerable groups.
Thank you for these insights. We added some additional detail throughout the conclusion section to address these comments while attempting to not overly increase the length of the article.
General comments
Comment 12. Linguistic review - I note that the term “Food In-security” is used, although the correct form is “Food Insecurity”. This is an error that requires correction, as the correct term is one recognized and used in the scientific literature and international policies.
Thank you for this observation. We believe this was a formatting issue and have ensured consistency throughout the document.
Comment 13. Clarity and conciseness - Some paragraphs are redundant and could be simplified to improve the flow of the text.
Addition of study limitations - A section dedicated to limitations would strengthen scientific rigor.
Thank you for this observation. We have done a full review of the article given the significant changes we implemented based on the excellent feedback we received. We did change the order and flow of some content and edited to simplify. Thank you again for your insights and suggestions.

Round 2
Reviewer 1 Report
Comments and Suggestions for Authors
The authors have effectively addressed all my comments. However, one point remains: please provide a more detailed discussion of the pros and cons of this program compared to similar initiatives implemented in other regions. This additional detail could enhance the readers' understanding and perspective on the current program.
Author Response
Thank you for this thoughtful suggestion about including comparative analysis with similar initiatives.
Our manuscript currently references several relevant coalition approaches in the Introduction section (line 45 to 56), including Janosky et al.'s work with a multisector community coalition in Summit County, Ohio [15], Michigan's Good Food Charter [17], and South Dakota's Local Foods Collaborative [24]. These initiatives demonstrate how coalitions and the CI framework have been successfully applied to address complex community challenges, including food security. However, we acknowledge that direct comparisons of emergency food response programs during the early pandemic period (March-December 2020) are limited due to:
- Our focus on evaluating coalition formation and functioning during the first 9 months of emergency food response
- The fact that most published studies examine longer-term food security initiatives rather than rapid emergency response coalitions
As more studies of pandemic-era emergency food response coalitions become available, comparative analysis would be valuable for future research to better understand the relative effectiveness of different approaches to addressing food security among older adults during crises.
Reviewer 4 Report
Comments and Suggestions for Authors
The manuscript is not submitted with track changes. In these conditions, it is very difficult to analyze the improvements made by the authors, during a second review, even I already read the new version. The journal's requirements are clear and I believe that the editors also specifically requested the revision of the manuscript, with track changes. Anyway, that's it now. It is difficult to redo it now with track changes. But if the authors have done everything they wrote in the cover letter, even if they did not present it concretely, I believe that is enough. I ask the editor to decide whether or not to accept the work, even if I select "accept in present form"
Author Response
Thank you for this feedback regarding track changes. We apologize for not submitting the manuscript with track changes. We made this decision because the revision involved substantial changes (approximately 40% of the original manuscript) in response to the comprehensive reviewer feedback. We were concerned that such extensive track changes would make the document difficult to read and review effectively. Nevertheless, we understand that having track changes would have made it easier to identify specific improvements.
We confirm that all changes described in our cover letter have been implemented in the manuscript. For future reference, we will work to find the best way to clearly show revisions while maintaining readability.
We appreciate your understanding and willingness to consider the manuscript despite this limitation.